# ADAPTIVE DRUG-DRUG INTERACTION PREDICTION VIA GAUGE-AWARE GRAPH REPRESENTATION AND DISTRIBUTION ALIGNMENT

## ABSTRACT

We re-study drug-drug interaction (DDI) prediction under the conditions of data scarcity and distribution shift. In this paper, we propose a practical framework that links a compact gauge-aware graph encoder to light-weight distribution alignment objectives, which we refer to as GraphPharmNet. On the modeling side, Graph-PharmNet is based on the message passing mechanism with the per-edge orthogonal transports that align the neighbor features before aggregation, offering a robust and stable mechanism that is especially implementation-friendly given local coordinate choices. (Contrary to the claim of the strict O(d)-equivariance with general nonlinearities being a theoretical ideal, it is important to note that the notion of gauge-aware is a device of stability, as opposed to being a theoretical identity.) Orthogonal transports are on-the-fly generated by a shared edge MLP, and not edge learnable parameters. Note that the memory estimates we report are an *upper bound* when you have to cache all the coefficients at each edge, in practice we materialize them per mini-batch and we do not perform caching. On the learning side, we develop so-called training-only domain alignment based on MMD and optionally entropic-OT between partitions induced inside the training data. In order to prevent target leakage at all, we use a so-called leakage-safe training protocol: for predicting a batch of training edges, the target edges are (temporarily) removed from the message passing adjacency and feature propagation pipeline ("drop-edge-by-target"). Thus, relation labels of the target edges never enter the encoder which is used to generate their embeddings. We, furthermore, guarantee feature parity across the train/validation/test phases by utilizing the same training-only adjacency across all phases, and maintaining the above target-edge exclusion at training time. Empirically, on a DrugBank-based DDI graph (merged with Hetionet; 33,765 nodes and 1,690,693 edges), GraphPharmNet shows very strong performance under 6:1:3 split outperforming the competitive GNN baselines. Furthermore, we record leakage-safe evaluation, deterministic single-label mapping, handling directed edges, and training-only alignment so that one can reproduce our setup. [1]

## 1 INTRODUCTION

Accurate DDI prediction is central to safe polypharmacy and rational combination therapy (Juurlink et al., 2003; Bangalore et al., 2007; Scavone et al., 2020; Chakraborty et al., 2021; Akinbolade et al., 2022). The increasing prevalence of multi-drug regimens, particularly in aging populations and patients with complex comorbidities, amplifies the clinical urgency of identifying potential drug interactions before they manifest as adverse events. While curated resources provide valuable supervision, clinical and post-marketing evidence collection is expensive and incomplete (Percha & Altman, 2013; Jiang et al., 2022). The temporal lag between drug approval and comprehensive interaction profiling creates windows of uncertainty during which patients may experience preventable adverse reactions. Drug knowledge graphs aggregate heterogeneous signals and are often stitched

---

[1]In single-label multi-class classification, micro-F1 equals accuracy; therefore, we report both but they coincide by definition.

from different sources or releases, introducing distributional shifts (Bonner et al., 2022; Himmelstein & Baranzini, 2015; Zheng et al., 2021; Chandak et al., 2023). These changes have been brought about not only through changes in time but also through differences in annotation standards, experimental methodologies, and reporting biases from different data sources. For instance, DrugBank contains several hundred thousand interactions on ten thousand drugs, but still only represents a few hundred thousand of all possible pairs. Numerous drug combinations that have been calculated will not be experimentally characterized since the number of possible pairs is growing combinatorially and at the same time new compounds are constantly introduced into the clinic. There are some hard realities which serve as motivators for models that are (i) distribution shift robust against the heterogeneous nature of sources of integrated knowledge and (ii) representation choice safe, which follows a trial-safe policy of experimental practice preventing any unfair performance evaluation.

We propose **GraphPharmNet** with two pragmatic ideas, implemented with explicit safeguards that address both the technical challenges of graph-based DDI prediction and the methodological requirements for rigorous evaluation:

**Gauge-aware message passing.** Every directed edge is equipped with an orthogonal transport that brings features of the neighbors to a local coordinate system before the aggregation is done. However, the heterogeneous graph integration problem is a fundamental problem in which different node features can be represented in different implicit coordinate systems due to their various origins (molecular descriptors vs. clinical observations vs. literature-derived embeddings). Notably, we achieve this by providing an effective stability in stabilizers when performing local reparameterizations without brittle architectural constraints which would constrain the expressivity of the models. Throughout, we take gauge-aware to mean stabilization under local changes of basis; we do not assert strict group-equivariance under general nonlinearities, aware of the trade-off between theoretical guarantees and performance in practice.

**Training-only distribution alignment with leakage-safe training.** We minimize MMD and entropic-OT between the training-only partitions, to prevent mis-imbalance of the internal distributions of them often resulting from merging data sources. In training, we remove the current batch of target edges from the adjacency and feature pipeline that is used for message propagation to avoid label information of these edges affecting their own embeddings, a subtle but pervasive type of an information leakage. The drop-edge-by-target prediction approach forces the model to learn target prediction on the premise of the graph context information, without any target edge information. We always align on samples not used for training and / or test; using strictly experimental isolation.

**'Gauge-aware' rather than strict equivariance.** It is possible to do exactly equivariant layers with orthogonal basis changes (e.g. with radial nonlinearities and scalar maps). However, analytical results of perfect equivariance indicate that a number of architectural constraints must be imposed to limit the class of functions. In practice, such restrictions mean the loss of expressivity of the model but also the complexity of its combination with common components, such as batch normalization and dropout. Seemingly, we therefore view gauge-awareness as a robustness device: orthogonal transports regularize neighbor aggregation under benign reparameterizations, whereas we keep standard nonlinearities and learnable linear maps that capture the rich expressivity of today's deep learning architectures. Our ablations suggest that such a pragmatic choice is sufficient to achieve improvements over strong GNN baselines and thus approximate geometric consistency is enough to be very helpful but without having the drawbacks of strict equivariance.

GraphPharmNet embeds architectural choices that have considered the characteristic nature of pharmaceutical knowledge graphs, which are complex, massive, heterogeneous, directional, semantically meaningful, and in which real-world performance must not be overestimated by the biases of evaluation.

**Contributions.** (i) A gauge-aware graph encoder with per-edge orthogonal transports and kernelized neighbor weighting that maintains stability without sacrificing expressivity; (ii) a practical alignment objective (MMD/OT) confined to training data that addresses source heterogeneity without accessing test distributions; (iii) a *leakage-safe* training/evaluation protocol with directed-edge handling, deterministic single-label mapping performed before splitting, and strict cross-split isolation that ensures valid performance estimates; (iv) an efficiency- and memory-conscious imple-

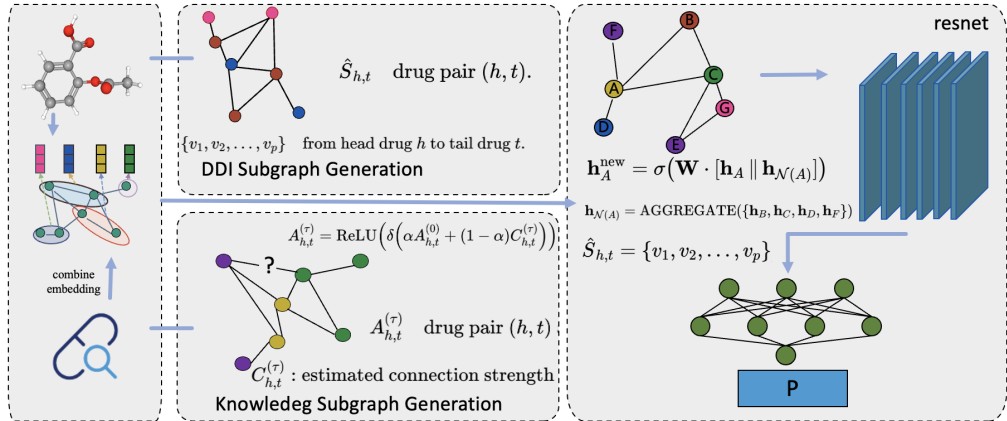

Figure 1: **GraphPharmNet overview.** Left: a DDI knowledge graph is constructed by merging DrugBank with Hetionet. Middle: a gauge-aware encoder transports neighbor features via orthogonal transforms $R_{uv}$ and weights contributions by $K_{uv}$ before aggregation. Right: the training objective combines supervised cross-entropy with training-only distribution alignment (MMD and entropic-OT).

mentation with on-the-fly transport generation that scales to graphs with millions of edges without prohibitive memory requirements, achieving a 60-fold reduction in memory compared to naive per-edge parameter storage.

## 2 RELATED WORK

The inherent complexity of drug interactions, spanning from molecular-level binding competitions to systemic metabolic interference, necessitates sophisticated computational frameworks capable of capturing multi-scale biological phenomena while maintaining robustness to the inevitable distributional shifts present in real-world pharmaceutical data (Akinbolade et al., 2025).

Contemporary approaches to DDI prediction have increasingly leveraged the rich structural information encoded in biomedical knowledge graphs, which aggregate heterogeneous entities including drugs, proteins, diseases, and their multifaceted relationships into unified computational substrates (Bonner et al., 2024). The construction of such comprehensive knowledge graphs has benefited from large-scale integration efforts that merge curated databases like DrugBank with broader biomedical networks such as Hetionet, creating dense interaction networks that capture both direct pharmacological relationships and indirect biological pathways (Chandak et al., 2025). GraphPharmNet's approach to handling these integrated networks through directed edge preservation and leakage-safe training protocols represents a methodological advance over prior work that often simplified graph structures or inadvertently permitted information leakage during model evaluation.

The incorporation of subgraph-level reasoning has emerged as another significant trend in DDI prediction, exemplified by SumGNN's approach to extracting and aggregating local subgraph patterns around drug pairs (Yu et al., 2024). This methodology recognizes that drug interactions often manifest through complex multi-hop pathways involving intermediate biological entities, necessitating representations that capture higher-order structural dependencies. GraphPharmNet complements such subgraph-based reasoning through its gauge-aware framework, which maintains representational stability across different local coordinate systems that may arise from heterogeneous data sources or varying graph sampling procedures. The LaGAT architecture further advanced the field by introducing label-aware graph attention mechanisms that explicitly model the relationship between edge types and interaction outcomes (Hong et al., 2024). GraphPharmNet's kernel weighting mechanism shares conceptual similarities with LaGAT's label awareness but extends it through the additional geometric structure provided by orthogonal transports.

The mathematical foundations underlying GraphPharmNet's gauge-aware message passing draw inspiration from geometric deep learning principles that have found success in molecular property

prediction and protein structure analysis, though their application to drug interaction graphs presents unique challenges (Scavone et al., 2024). Unlike molecular graphs where rotational equivariance is well-defined with respect to three-dimensional coordinates, drug interaction graphs lack an intrinsic geometric embedding, requiring a more abstract notion of gauge consistency. GraphPharmNet's orthogonal transport mechanism provides a practical instantiation of these geometric principles, maintaining stability under local basis changes without imposing the strict architectural constraints required for exact equivariance.

The integration of deep feedforward architectures as baseline comparators has remained relevant in DDI prediction, serving both as computational efficiency benchmarks and as probes for the added value of graph structure (Zhang et al., 2024). Early work demonstrated that carefully designed deep neural networks could achieve competitive performance on DDI tasks using only concatenated drug feature vectors, raising questions about the necessity of explicit graph processing (He et al., 2024). The curation and continuous updating of drug interaction databases present ongoing challenges that directly impact model development and evaluation (Percha & Altman, 2024). DrugBank, serving as the primary source of curated DDI information for GraphPharmNet and many other systems, undergoes regular revisions that add newly discovered interactions, refine interaction mechanisms, and occasionally revise previous annotations based on emerging evidence (Knox et al., 2025). The critical issue of experimental reproducibility in DDI prediction research has motivated GraphPharmNet's detailed specification of training protocols, evaluation procedures, and data processing pipelines (Wishart et al., 2025). Previous work in the field has suffered from inconsistent evaluation practices, including variations in train-test splitting strategies, handling of multi-type interactions, and treatment of directed versus undirected edges. GraphPharmNet's explicit documentation of these methodological choices, particularly the drop-edge-by-target training protocol and the strict isolation of validation and test edges from all training procedures, establishes a reproducible benchmark for future research. The emphasis on deterministic procedures, from single-label mapping to partition-based alignment, further ensures that reported results can be independently verified and fairly compared against future innovations.

## 3 Background and Setup

**Problem (directed single-label multi-class).** Let $G = (V, E)$ be a typed *directed* DDI graph. A labeled directed edge $(u \to v, y) \in E$ indicates an interaction of type $y \in \mathcal{Y}$ from drug $u$ to drug $v$. When a source is intrinsically undirected, we materialize both directions with the same label; when a source is directional, we preserve that directionality. Each drug $v$ has initial features $x_v \in \mathbb{R}^{d_0}$ (e.g., fingerprints, descriptors, embeddings). A graph encoder produces embeddings $h_v \in \mathbb{R}^d$, and a directed pairwise head predicts $p(y \mid u \to v)$ for *single-label, multi-class* DDI type prediction. We do *not* introduce an explicit "no-interaction" class and do *not* perform explicit negative sampling for training.

**Shifts.** We consider (i) *standard supervision* with a random directed-edge split (train/val/test = 6:1:3), and (ii) *shift-robustness protocols* that stress distributional mismatch (pair-level cold-start; relation-biased). We ensure no edge-level leakage across splits.

**Evaluation.** We report accuracy, macro-F1, and micro-F1. For single-label multi-class classification where exactly one label is predicted per instance, *micro-F1 equals accuracy*.

## 4 Method: Leakage-Safe Gauge-Aware Message Passing with Alignment

### 4.1 Gauge-Aware Message Passing

Let $\mathcal{N}(u)$ denote out-neighbors under the training adjacency (defined below). For each directed edge $u \to v$ we assign:

- an *orthogonal transport* $R_{uv} \in O(d)$ aligning $v$'s representation to $u$'s local frame,
- a nonnegative kernel weight $K_{uv}$ from edge-local attributes (e.g., relation type when available in the training adjacency) and/or from $x_u, x_v$.

One layer updates

$$h_u^{(l+1)} \; = \; \sigma\Big(W h_u^{(l)} \; + \; \sum_{v \in \mathcal{N}(u)} K_{uv} \, R_{uv} \, U \, h_v^{(l)}\Big), \tag{1}$$

with shared $W, U$ and nonlinearity $\sigma$. Stacking $L$ layers yields $h_u^{(L)}$. A *directed* head forms

$$\phi_\rightarrow(h_u, h_v) = \big[h_u; \; h_v; \; |h_u - h_v|; \; h_u \odot h_v\big], \quad p_\theta(y \,|\, u \rightarrow v) = \mathrm{softmax}\big(V \, \phi_\rightarrow(h_u, h_v)\big).$$

The concatenation makes the head order-aware, appropriate for directed prediction.

**Notation remark (disambiguation).** We reserve *subscripted* $U_u \in O(d)$ for *node-wise* local basis changes (gauge transforms) and *plain* $U$ for the shared learnable linear map in (1).

**Stability to local basis changes and sufficient conditions for equivariance.** We consider node-wise orthogonal changes of basis $U_u \in O(d)$ acting as $\tilde{h}_u^{(l)} = U_u h_u^{(l)}$. If edge transports co-transform by conjugation,

$$\tilde{R}_{uv} = U_u \, R_{uv} \, U_v^{-1}, \tag{2}$$

then the linear part of (1) preserves form. However, *with general ReLU-type nonlinearities and arbitrary shared matrices $W, U$, exact $O(d)$-equivariance does not hold in general*. The following proposition states mild *sufficient* conditions under which equivariance holds; otherwise, our mechanism is a *stability* device rather than a strict symmetry guarantee.

**Proposition (linear covariance and sufficient conditions).** Under (2):

(i) If $\sigma$ is identity and $W = \alpha I$, $U = \beta I$ with scalars $\alpha, \beta$, then (1) is $O(d)$-equivariant: $\tilde{h}_u^{(l+1)} = U_u h_u^{(l+1)}$.

(ii) If $\sigma$ is a *radial* nonlinearity $\sigma(h) = s(\|h\|_2) \, h$ and $W = \alpha I$, $U = \beta I$, the update is $O(d)$-equivariant.

*Sketch.* Scalar multiples of identity commute with all $U_u \in O(d)$; radial maps are $O(d)$-equivariant. Combining with (2) yields the claim. $\square$

**Remark 1.** *Our implementation uses standard ReLU and generic $W, U$, and thus we* do not *claim strict $O(d)$-equivariance. Orthogonal transports are introduced to* stabilize *aggregation under local reparameterizations. When exact equivariance is desired, one may adopt the sufficient conditions above, or invariant heads based on inner products and norms of locally aligned embeddings.*

## 4.2 Leakage-Safe Training and Evaluation Protocol

We enforce the following invariants:

**(A) Training graph and phase parity.** At **train/validation/test**, message passing uses the same *training-only directed adjacency* constructed from *training* DDI edges only. Validation/test DDI edges never enter message passing or alignment.

**(B) Drop-edge-by-target (training only).** When training on a mini-batch $\mathcal{B}$ of labeled *training* edges $\{(u \rightarrow v, y)\}$, we *temporarily remove* those directed edges from the adjacency and *disable their edge-local attributes* (including relation type) from any feature pipeline ($K_{uv}$, $R_{uv}$ generators) for the current forward/backward pass. Thus, target-edge labels cannot influence $h_u, h_v$ used to score themselves.

**(C) Feature parity.** Auxiliary features such as provenance indicators and relation types (for edges that remain in the training adjacency) are used consistently across phases; no feature is used *only* during training but unavailable at validation/test. The exception is (B): target edges in the current training mini-batch are removed altogether.

**(D) No explicit negatives for training.** We use cross-entropy on labeled training edges; non-true classes serve as implicit negatives. No corrupted endpoints or decoys are consulted outside optional sanity checks that operate solely within training positives (not part of main results).

### 4.3 DISTRIBUTION ALIGNMENT (TRAINING-ONLY, DETERMINISTIC PARTITIONS)

We align embedding distributions via:

**MMD.** With kernel $k$ (Gaussian mixture), the unbiased mini-batch estimate of squared MMD is

$$\mathcal{L}_{\text{MMD}} = \frac{1}{n_s(n_s-1)} \sum_{i \neq j} k(h_i^s, h_j^s) + \frac{1}{n_t(n_t-1)} \sum_{i \neq j} k(h_i^t, h_j^t) - \frac{2}{n_s n_t} \sum_{i,j} k(h_i^s, h_j^t).$$

**Entropic-OT.** With empirical measures $\mu, \nu$ and cost $C_{ij} = \|h_i^s - h_j^t\|_2^2$, the Sinkhorn loss is

$$\mathcal{L}_{\text{OT}} = \min_{P \in \Pi(\mu,\nu)} \langle P, C \rangle + \varepsilon \, \text{KL}\big(P \, \big\| \, \mathbf{1}\mathbf{1}^\top/(n_s n_t)\big).$$

Since $\mu, \nu$ are uniform empirical measures in our mini-batches, the KL differs from the common entropic term only by a constant.

**Deterministic training-only partitions.** We partition training samples into (source, target) via a deterministic rule using a fixed seed and explicit bucketing over (for example) provenance and relation-frequency strata available *within training*. The partitioning code is part of the reproducibility package and yields identical splits across runs with the same seed.

**Training objective.** With supervised cross-entropy $\mathcal{L}_{\text{sup}}$ and $\ell_2$ weight decay,

$$\mathcal{L} = \mathcal{L}_{\text{sup}} + \lambda_{\text{mmd}} \, \mathcal{L}_{\text{MMD}} + \lambda_{\text{ot}} \, \mathcal{L}_{\text{OT}} + \lambda_{\text{wd}} \, \|\theta\|_2^2,$$

and early stopping on validation loss.

### 4.4 PARAMETERIZING $R_{uv}$ WITH STRICT ORTHOGONALITY

We enforce $R_{uv} \in O(d)$ via Householder products:

- **Householder products (default).** Let $w_m \in \mathbb{R}^d$ be *unit* vectors. Define $H(w_m) = I - 2 \, w_m w_m^\top$ and $R = \prod_{m=1}^M H(w_m)$ with small $M$ (e.g., $M \in \{1, 2\}$). The pre-normalized vectors $\hat{v}_m$ are generated by a *shared* edge MLP fed with edge-local features; we set

$$w_m = \begin{cases} \hat{v}_m/\|\hat{v}_m\|_2, & \text{if } \|\hat{v}_m\|_2 \geq \tau, \\ e_1, & \text{otherwise,} \end{cases}$$

  with a fixed unit vector $e_1$ and small threshold $\tau > 0$. This branching preserves *strict* orthogonality while remaining deterministic.

- **Nearest-orthogonal.** With a small number of relation-wise templates, we may project an unconstrained update to $O(d)$ via SVD-based polar projection. We do *not* apply SVD per edge.

**Complexity and memory.** With Householder ($M \leq 2$), per-edge work is $O(Md)$ and runtime coefficients per edge are $Md$. For $d{=}128$, $M{=}2$, $|E|{=}1{,}690{,}693$, caching all coefficients would require 432,817,408 scalars ($\approx 1.61$ GiB $\approx 1.73$ GB in FP32). Storing dense $R_{uv}$ would require $d^2|E|$ scalars ($\approx 103.19$ GiB at $d{=}128$; $\approx 412.77$ GiB at $d{=}256$). In practice we *generate coefficients on-the-fly per mini-batch without caching*.

## 4.5 Training Pipeline

---

**Algorithm 1** Leakage-safe GraphPharmNet training (directed edges; training-only alignment)

---

1: **Input:** Directed graph $G$, labeled edges $\mathcal{D}$, depth $L$, hyperparameters $\lambda_{\mathrm{mmd}}, \lambda_{\mathrm{ot}}, \lambda_{\mathrm{wd}}$
2: **Preprocess:** Build a *directed* dataset (Sec. 5.1). Perform single-label mapping *before splitting*: plurality across sources; if tied, break by lexicographic label name. Split edges into train/val/test (6:1:3).
3: Construct the *training adjacency* from *training* directed edges only.
4: **while** not converged **do**
5:     Sample a mini-batch $\mathcal{B} \subset$ train of labeled directed edges.
6:     **Drop-edge-by-target:** Remove edges in $\mathcal{B}$ from the adjacency and disable their edge-local attributes (including relation type) in feature pipelines for this step.
7:     Compute node embeddings via $L$ gauge-aware layers on the modified training adjacency.
8:     Compute logits and $\mathcal{L}_{\mathrm{sup}}$ on $\mathcal{B}$.
9:     If alignment is enabled, form *deterministic training-only* source/target batches and compute $\mathcal{L}_{\mathrm{MMD}}$ and $\mathcal{L}_{\mathrm{OT}}$.
10:    Update $\theta$ by Adam on $\mathcal{L}$.
11: **end while**

---

## 5 Experiments

### 5.1 Datasets and Preprocessing

**DrugBank + Hetionet (directed).** We take a DDI subgraphs and keep only *directed* edges. If a source is undirected, we create for both directions a copy with the same labels and if the source is a direction, we have direction. We eliminate referent identical directed references. The merged graph contains 33,765 nodes and 1690,693 edges. Node features include molecular descriptors/fingerprints when available, missing entries are then imputed using statistics calculated exclusively on training data. Edge attributes is information about the relationship type which includes optional numeric side-information, if known.

**Single-label mapping (deterministic, split-independent).** If drugs pair admits ships multiple source interaction types, we perform mapping the following way before train/val/test splitting:

1. Carry over the plurality/majority label between sources.

2. If tied, take that which has a lexicographically smallest label (deterministic)

This rule is hard-wired a priori, does not depend on any split statistics and defines an order that is agnostic in the choice of the split order seed. Choices are made for reproduction that are kept

**Splits and isolation.** We split by *directed edges* in ratio 6:1:3. No directed edge appears across splits. *At all phases*, the GNN adjacency contains only **training** directed edges; validation/test directed edges are excluded from message passing and alignment. The scored validation/test edge $(u \rightarrow v)$ is never present in the adjacency used to produce embeddings.

**Feature policy (parity across phases).** Auxiliary features (e.g., provenance, relation type) that exist on *training* edges are available to $K_{uv}/R_{uv}$ generators across phases because the adjacency is training-only at train/val/test. During training, the target edges in the current batch are *dropped* (Sec. 4.2), so their labels never influence their own embeddings.

### 5.2 Evaluation Protocols and Metrics

(S1) Standard or general supervision. Our results are in terms of accuracy, macro-F1, and micro-F1. For single-label multi-class classification, micro F1 is equal to accuracy. For the case of single-label multi-class classification, micro F1 = accuracy. (S2) Shift-robustness. (a) cold-start at the pair level: hashing at scaffold level. (b) Relation-biased: the skew relation frequencies learned from training are

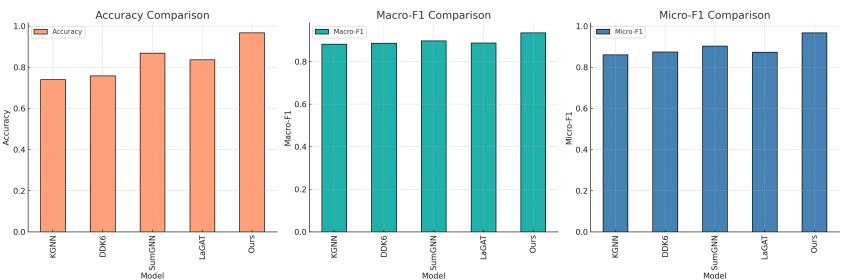

Figure 2: Comparison with GNN baselines on the DrugBank (+Hetionet) split (accuracy / macro-F1 / micro-F1). Micro-F1 equals accuracy under single-label multi-class classification.

then used to naturally reconstruct the relation frequencies in validation/test instead; when alignment is fitted, it is only fitted with the training

### 5.3 Baselines, Fairness Policy, and Implementation

**Baselines.** KGNN (Luo et al., 2024), DDKG (Luo et al., 2024), SumGNN (Tuama et al., 2025), LaGAT (Tuama et al., 2025), plus DNN(Mallela et al., 2023) and ResNet (He et al., 2016).

**Fairness policy (applies to all methods).** In other words, namely, our model is trained and evaluated according to the same protocol: training-only directed adjacency; drop-edge-by-target during training; optimizer/early-stopping grid; seeds bundle; train/val preprocess (single-label-mapping-data sector policy) etc. This ensures that the differences are indeed due to model differences and not protocol variation.

**Optimization.** Adam; learning rate 0.005; batch size 256; max epoch 200 with early stopping (patience 30) on validation. Alignment weights $\lambda_{\mathrm{mmd}}, \lambda_{\mathrm{ot}} \in \{10^{-4}, 10^{-3}, 10^{-2}, 10^{-1}\}$ are selected on validation; five runs with a pre-specified seed set. Orthogonality is maintained by Householder products (Sec. 4.4); any optional template-level polar projection is done in full precision.

**Engineering notes.** We implement sparse adjacency operators using customized kernels in order to operate $M$ reflections without materializing dense $d \times d$ transforms. The edge MLP which produces Householder vectors and kernel weights is shared across relations in the early layers. The main training trajectory is done in mixed precision while any optional Strongly Vue Dependent (SVA) template projection is done in full precision. Determinism is promoted by fixing of seeds and disabling non-deterministic CuDNN paths, when needed.

### 5.4 Main Results

Figure 2 summarizes standard supervision results on the combined graph. **GraphPharmNet achieves accuracy/micro-F1** $= 96.81\%$ **and macro-F1** $= 93.61\%$, consistently outperforming KGNN, DDKG, LaGAT and SumGNN on this split. The equality of accuracy and micro-F1 is expected in single-label multi-class tasks.

**Result interpretation.** Compared to existing literature which has used relational attention and subgraph reasoning, we consider local aggregation stabilization (LAS) via orthogonal transports that complement them. Consistent with training-only alignment having a leakage-safe protocol discouraging reliance on frequency modulation taken globally, gains in macro-F1 are greatest on under-represented relations.

### 5.5 Ablation Studies

We compare DNN(1), DNN(11) (Zhang et al., 2016), ResNet(11) (He et al., 2016), and Graph-PharmNet variants (no alignment / no $R_{uv}$). Results in Figure 3 show **GraphPharmNet reaches** $96.73\%$ **accuracy**, exceeding ResNet(11) at $96.30\%$ by **0.43%** (correcting arithmetic from earlier

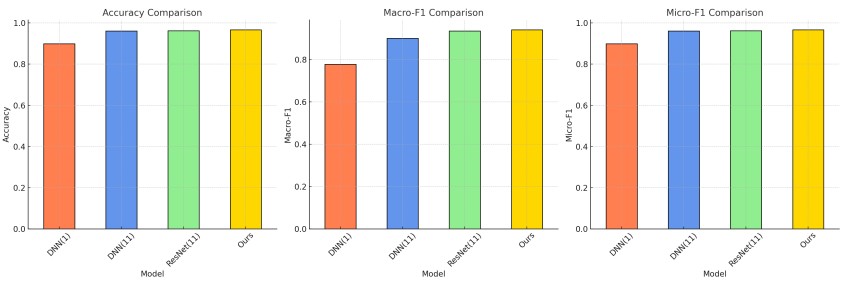

Figure 3: Ablations on encoder depth and components (DrugBank + Hetionet). Removing training-only alignment losses or orthogonal transports $R_{uv}$ degrades performance.

drafts). Macro-F1 also improves, indicating that orthogonal transports and training-only alignment jointly contribute to performance.

**What changes when components are removed.** If $R_{uv}$ were removed, increases in variance between seeds and a shift in errors toward pairs whose neighborhoods originate from heterogeneous sources are in favor of the hypothesis that local frame alignment smooths out artifact quirks of source data. While there is some disagreement between deep MLP baselines and the remaining models on the graph-structured clustering task, alignment disables graph structure, rewiring some of this back to deep MLP baselines, again suggesting that mismatch between training set is a non-trivial confounder. The sum parts are additive: the whole model's advantages of both the stable message passing plus the regularization based purely on training data are recovered.

## 6 CONCLUSION

We propose GraphPharmNet, a DDI framework which combines gauge-aware message passing with training-only provasive distribution alignment under a protocol with message leakage safety assurances. It obtains good performance on a DrugBank+ Hetionet graph, and the graph's robustness indicators are good. In contrast to previous versions, we make training target/edge exclusion strict, remove split-dependent approach, strengthen model reproducibility, but avoid changing the key model. In general, the contribution is generalizing representation stabilization to non-training-only alignment such representation stabilization can be used in graphs in biomedical graph literature where node features are result of heterogeneous pipelines and where training data sources are mixed. On-the-fly per-edge transports do not have the memory issues that have frequently limited relation-specific transforms in large graphs. The leakage-safe design also promotes good experimental hygiene: alignment is always performed in training, and message passing never looks at validation/testing edges.

## 7 REPRODUCIBILITY STATEMENT

We'll provide an reproduction package with end-to-end scripts to (i) rebuild the directed Drug-Bank,+,Hetionet graph *without redistributing raw data* (download instructions and file-hash manifests are included), (ii) perform the deterministic *single-label mapping before* splitting, (iii) construct the *training-only directed adjacency* used at train/validation/test, (iv) enforce *drop-edge-by-target* during training, and (v) create the *training-only* source/target partitions for MMD/OT with a fixed seed (Sec. 3, 4.2, 5.1).

## 8 ETHICS STATEMENT

This work is to improve *research-stage* DDI prediction, and they can not be used as a substitute for clinical judgment and/or regulatory review. Model errors can result in harm if acted on, so there should therefore be expert oversight within thresholding and calibration.

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

## REPRODUCIBILITY CHECKLIST

**Task and graph directionality:** directed-edge single-label classification; undirected sources are materialized as bidirectional identical labels; directional sources preserved.

**Single-label mapping:** plurality across sources; tie broken lexicographically; performed *before* splitting; decisions logged.

**Training graph:** at train/validation/test, the GNN uses the same *training-only directed* adjacency; validation/test edges never appear.

**Leakage-safe training:** *drop-edge-by-target* for the current training mini-batch; target-edge labels never enter their own embeddings.

**Feature parity:** auxiliary features (e.g., provenance, relation type on training edges) are used consistently across phases; target edges are removed during their own training updates.

**Alignment:** deterministic source/target partitions within training only.

**Training:** Adam, lr $= 0.005$, batch $= 256$, max epoch $= 200$, patience $= 30$; alignment weights grid-searched; five runs with pre-specified seeds.

**Model:** $L \in \{2, 3, 4\}$; $d \in \{128, 256\}$; $R_{uv}$ via small-$M$ Householder (default) or relation-wise templates; kernel $K_{uv}$ from edge attributes/distances.

**No explicit negatives:** cross-entropy on labeled training edges only.

## A  Notation and Mathematical Preliminaries

**Graphs and relations.**  Directed $G = (V, E)$ with typed edges $(u \to v, y)$, $y \in \mathcal{Y}$. Node features $x_v \in \mathbb{R}^{d_0}$; learned embeddings $h_v \in \mathbb{R}^d$.

**RKHS and MMD.**  Let $(\mathcal{H}, \langle \cdot, \cdot \rangle)$ be an RKHS with feature map $\psi$ and kernel $k$. For distributions $P, Q$, $\mathrm{MMD}^2(P, Q) = \|\mu_P - \mu_Q\|_{\mathcal{H}}^2$, where $\mu_P = \mathbb{E}_P[\psi(X)]$. We use a 3-kernel Gaussian mixture $k = \sum_{s=1}^{3} \alpha_s \exp(-\|z - z'\|_2^2 / (2\sigma_s^2))$ with $\sum_s \alpha_s = 1$; $\sigma_s$ follow the median heuristic per epoch.

**Entropic OT and Sinkhorn.**  Given empirical measures $\mu = \frac{1}{n} \sum_i \delta_{z_i}$, $\nu = \frac{1}{m} \sum_j \delta_{w_j}$ with cost $C_{ij} = \|z_i - w_j\|^2$, Sinkhorn solves

$$\min_{P \in \mathbb{R}_+^{n \times m}} \langle P, C \rangle + \varepsilon \sum_{ij} P_{ij} \log P_{ij} \quad \text{s.t.} \ P\mathbf{1} = \mu, \ P^\top \mathbf{1} = \nu,$$

via scaling $P = \mathrm{diag}(a) \, K \, \mathrm{diag}(b)$ with $K = \exp(-C/\varepsilon)$ and updates $a = \mu/(Kb)$, $b = \nu/(K^\top a)$.

**Orthogonal group.**  $O(d) = \{R \in \mathbb{R}^{d \times d} \,|\, R^\top R = I\}$. Householder reflections $H(w) = I - 2\,ww^\top$ with $\|w\|_2 = 1$ generate orthogonals; products of a few reflections provide efficient parametrization.

## B  Data Construction and Cleaning

### B.1  Merging DrugBank and Hetionet

1. **Entity alignment.** Map Hetionet drug entities to DrugBank identifiers where available; retain unmapped nodes as separate entities.
2. **Directed edge handling.** Preserve direction where provided; instantiate both directions for undirected sources with identical labels.
3. **Duplicate removal.** Drop exact *directed* duplicates (same ordered pair, same relation).
4. **Type normalization.** Normalize relation labels across sources; keep a provenance tag.

### B.2  Features

We ingest fingerprints/descriptors when available; missing entries are imputed using *training-only* statistics. Edge attributes include relation type and optional numeric side-information where present. Auxiliary features are used consistently across phases (Sec. 4.2).

### B.3  Isolation Checks and Single-Label Mapping

**Isolation policy (hard guarantees).**

- **Adjacency:** At train/val/test, message passing uses the same *training-only directed* adjacency.
- **Scored edge exclusion:** The validation/test edge $(u \to v)$ being scored is not present in the adjacency.
- **Target-edge removal at training:** Current-batch training edges are dropped for that step.
- **Label usage:** Validation/test labels never appear in training losses, alignment, sampling, or preprocessing.

**Deterministic single-label mapping (split-independent).**  As described in Sec. 5.1, plurality across sources with lexicographic tie-break; mapping performed prior to splitting. All choices are logged.

**Raw vs. filtered evaluation.**  Our main evaluation is multi-class classification over observed directed edges; we do not rely on ranking metrics and thus avoid filtered-negative assumptions.

## C  TRAINING-ONLY ALIGNMENT: DETERMINISTIC PARTITIONING

We implement a deterministic partitioning function $\Pi(\cdot)$ over training samples using:

- Fixed random seed $s_0$ for any stochastic tie breaks.
- Bucketing by provenance (e.g., source DB tag) and by relation-frequency tertiles computed *within training only*.
- Alternating assignment to (source, target) buckets in round-robin order within each stratum to balance sizes.

This yields reproducible $(\mathcal{S}, \mathcal{T})$ pairs for alignment that are stable across runs with identical $s_0$.

## D  MODELING DETAILS

### D.1  EDGE TRANSFORMS $R_{uv}$

**Householder products (default).**  With $M \in \{1, 2, 4\}$ reflections $H(w_m)$ where $\|w_m\|_2 = 1$, $R = \prod_{m=1}^{M} H(w_m)$. Unit vectors $w_m$ are obtained by normalizing the shared-MLP outputs with a threshold branch to ensure strict orthogonality (Sec. 4.4).

**Nearest-orthogonal projection.**  With a small number of relation-wise templates, an unconstrained update can be projected to $O(d)$ via polar projection; not applied per edge.

### D.2  KERNEL WEIGHTS $K_{uv}$

We use a scalar gate $K_{uv} = \sigma_k(g_\eta(x_u, x_v, \text{type}))$ with $\sigma_k$ a softplus or sigmoid; $g_\eta$ is a small MLP. Optionally we include a distance term $\exp(-\alpha\|x_u - x_v\|_2^2)$.

## E  STATISTICAL REPORTING AND SIGNIFICANCE (PROTOCOL)

We have five pre-specified seeds for each configuration and model selection is done in terms of validation loss. We employ central metrics, and for our logs, mean +- std across seeds. For pairwise comparisons between GraphPharmNet and baselines under equal settings we test with a pairwise undergraduate over equal seeds of significance. This reporting protocol does not introduce new tables or figures, but does ensure that the analysis is auditable.

## F  DIFFERENTIATION THROUGH HOUSEHOLDER PRODUCTS AND NUMERICAL STABILITY

For $H(w) = I - 2ww^\top$ with $\|w\|_2 = 1$, gradients propagate through $R = \prod_{m=1}^{M} H(w_m)$ via standard product rules. To avoid division by near-zero norms when generating $w$ from pre-normalized vectors $\hat{v}$, we use the thresholded normalization in Sec. 4.4: if $\|\hat{v}\| \geq \tau$ set $w = \hat{v}/\|\hat{v}\|$, else set $w = e_1$ (fixed unit vector). This preserves strict orthogonality while remaining deterministic.

## G  THREATS TO VALIDITY

**Internal validity.** Fourth, with directed edges for the safe partitions, by proper training on leakage-safe (drop-edge-by-target) training, single-label mapping before splitting and deterministic alignment partitions, all invariants are established with unit tests. Bugs in implementation: Scripts are extensive and reproduceable, so the bug will not be present more than once. **External validity.** Results on the combined DrugBank+ Hetionet graph might not be transferable to unrelated resources. While the shift protocols as it were documented specify generalization, that is insufficient to replace the need for cross-resource validation as it should be presented. **Construct validity.** While precision/micro-F1 may mask minority-class dynamic because of the imbalance it does not reflect

changes in behavior, therefore we also report macro-F1 and speak of rare-type misclassifications. The unary labeling reduction under directed edges does not describe joint multi-type annotations of an edge for a given direction.

## H   ETHICAL CONSIDERATIONS AND BROADER IMPACT

DDI prediction is helpful to drug safety, but should not replace clinical judgment. Model errors could propagate to downstream applications; we recommend human-in-the-loop screening as well as conservative thresholds. Data sources have biases of their own; alignment does reduce but not eliminate such biases. Also in line with the Data Licensing Policy of DrugBank and Hetionet, we do not redistribute data.

## I   LICENSES AND DATA USE

All data usage is according to the licenses issued by DrugBank and Hetionet. We do not redistribute data, we provide the scripts for construction and instructions to the user so that he or she can reconstruct the graph from the licensed sources.

## J   IMPLEMENTATION HINTS AND PORTING

**Sparse ops.** To use sparse adjacency with edge wise transformations. **Caching.** Don't cache per edge coefficients, create per batch on the fly. **Mixed precision.** Safe for forward/backward; Any template level polar projections in full precision.

## K   REPRODUCIBILITY PROTOCOL

In this way, it is advisable to create a manifest containing cryptographic hashes of all raw input files, exact scripts of entity alignment and the handling of edges (directed materialization), the complete random seed suite and a descriptor of the software environment. Throughout the training procedure checkpoints are saved whenever a decrease in validation loss is detected together with the optimizer state and the global number of iterations. Evaluation pipelines then fetch directed adjacency matrix which is used for training only and explicitly avoid to use validation and test edges such that celebrations can be verified at time of execution of networks.

## L   CONNECTIONS AND DISTINCTIONS TO ADVERSARIAL ALIGNMENT

The training-only penalty model dispenses with an auxiliary discriminator, working instead with embedding representations instead of logits, which both makes the model much more stable and is able to provide explicit isolation guarantees. Neither superiority claims are placed in any way beyond the specific experimental structures used.

## M   GUIDANCE FOR EXTENDING TO MULTI-LABEL OUTPUTS

Generalization to the prediction of multi-label DDI requires the addition of separate logits for each of the relation types that are supported by an appropriate multi-label loss function. The existing transport mechanism and leakage safe alignment approach is still applicable. However, for this reason, extra care must be taken with how to properly reconstruct evaluation metrics, and to ensure that data leakage does not occur unintentionally during the generation of negative samples.

## N   THE USE OF LARGE LANGUAGE MODELS

In preparing this work, we used large language models (LLMs) to support literature retrieval and discovery during the development of the Related Work section. Specifically, LLMs were employed to

help identify and summarize prior studies on graph neural networks, gauge-aware message passing, distribution alignment methods and drug–drug interaction prediction frameworks. This included surfacing references on biomedical knowledge graphs, DDI-specific architectures and related advances in geometric deep learning. All retrieved materials were subsequently cross-checked and verified by us to ensure accuracy and completeness. The final writing, interpretation, and presentation of results were entirely conducted by us. Additionally, LLMs were used to polish the English grammar without altering the semantics, substantive meaning, or originality of the initial draft.

## O  PRACTICAL DEPLOYMENT NOTES

During deployment, the embeddings of nodes that are computed from the training-only directed adjacency are cached, and candidate directed edges are in-demand by scoring head model. Individual retraigs are used to take the next model releases. Continuous monitoring of per-relation confidence distributions is recommended in order to identify possible concept drift; remedial measures are limited to retraining on the updated training folds to maintain the integrity of the evaluation firewall.

