# OpenReview forum: "Adaptive Drug-Drug Interaction Prediction via Gauge-Aware Graph Representation and Distribution Alignment"
_ICLR.cc/2026/Conference — ICLR 2026 Conference Desk Rejected Submission_

### Official Review · Reviewer_naJm · 2025-10-18

**Soundness:** 1
**Presentation:** 1
**Contribution:** 1
**Rating:** 0
**Confidence:** 3

**Summary:**

The paper aims to revisit the DDI prediction problem under data scarcity and distribution shift. It introduces GraphPharmNet, a framework combining a gauge-aware graph encoder with lightweight distribution alignment strategies. The work claims novelty in using per-edge orthogonal transports (computed via a shared edge MLP) to align neighborhood features before aggregation, promoting stability without requiring strict O(d)-equivariance. On the training side, the authors aims to design a leakage-safe protocol with training-only domain alignment (using MMD and optional entropic-OT), ensuring fair evaluation and reproducibility. Experiments on a large merged DrugBank–Hetionet dataset show consistent gains over baseline GNNs (though many SOTA models are excluded from the comparison), though the approach’s scalability and computational trade-offs warrant closer scrutiny.
Ovwerall, I think, this paper presents an interesting idea and a commendable effort, but the clarity and coherence of presentation are weak, and the experiments do not sufficiently support the claims. Overall, it lacks sound motivation, depth, and analytical rigor. Strengthening the structure, related work, and experimental validation would greatly improve its suitability for ICLR.

**Strengths:**

1. Introduces a gauge-aware graph encoder that integrates orthogonal transport mechanisms in message passing, offering a fresh perspective on stabilizing feature aggregation in molecular graphs.
2. Their proposed leakage-safe training protocol and training-only distribution alignment demonstrate careful attention to experimental rigor and reproducibility, which are often overlooked in DDI prediction research.
3. Aims to address practical challenges of distribution shift and data scarcity in biomedical graphs, an underexplored yet important direction for improving real-world drug–drug interaction prediction.

**Weaknesses:**

1. The problem doesn't seem well-motivated or some related literature is missing. The discussion contains several problems in current DDI works, but their approch and modeling choices, doesn't seem to have any relation to these.
2. Informaiton in lines 069-097 seems out of context without any references and problem formualtion; thus very hard to understand and relate.
3. Several relevant DDI works are missing from the discussion. DOIs: 10.1145/3511808.3557648, 10.48550/arXiv.1905.00534, 10.1093/bib/bbab441, 10.48550/arXiv.2403.17210, 10.1093/bib/bbab133, 10.1093/bib/bbac597, 10.48550/arXiv.2209.09941, 10.1039/D2SC02023H, 10.48550/arXiv.2508.06576.
4. In Figure 2 and 3, mention the exact values, increase font sizes as it is not mreadable in current format. Tables are preferred.
5. In Section 5.3 : Baselines, it is not clear why you didn't compare it with existing GNN-based DDI models (DOIs: 10.1145/3511808.3557648, 10.48550/arXiv.1905.00534, 10.1093/bib/bbab441, 10.48550/arXiv.2403.17210, 10.1093/bib/bbab133, 10.1093/bib/bbac597, 10.48550/arXiv.2209.09941, 10.1039/D2SC02023H)? Also, connecting to Figure 2, exisitng DDI models is reported to gain 90-98% accuracy, 99% AUROC in 2023. So, a discussion is expected on how this work differencites from them in terms of application and provides new insights to DDI researchers.
6. Results section is very narrow and do not justify a lot of claimed contributions. A lot of theoretical contributions are made, but the result section do not reflect how these things are actually making a real impact in the modeling; or, how these addresses a practical problem in DDI.
6. Ablations are not clear. Also, more detailed ablaitons are expected, concerning the contribution of the paper. For example, study the impact of *gauge-aware*ness, the alignment objective (MMD/OT), and more. A lot of things are introduced, but their impact or necesity was not establsihed proeprly.
7. Contributions state *an efficiency- and memory-conscious implementation with on-the-fly transport generation that scales to graphs with millions of edges without prohibitive memory requirements*, but there is not scaling studies presented.
8. Presentation and writing quality is poor. Very hard to foillow along and understand. Paper structure, word choices and content flow should be improved.

**Questions:**

1. The section in lines 044-060 seems disconnected to 060-064. Explain in details both motivators you mentioned, *(i) distribution shift robust against the heterogeneous nature of sources of integrated knowledge and (ii) representation choice safe*. I didn't find any prior discussion to *distribution shift*, what it is in the context of DDI? how and why it happens? and how and why it matters? Also, the second point is unclear too. What do you meant by "*whichfollows a trial-safe policy of experimental practice preventing any unfair performance evaluation.*"? As per current information, I don't see these methodological requirements being established properly.
2. In Line 069, before discussing the technical terms like *edge*, maybe the graph and problem preliminaries should be introduced; so that the readers can understand the situation better.
3. L70/71 says,*the heterogeneous graph integration problem is a fundamental problem* - any related works or references?
4. What does it mean by *we achieve this by **providing an effective stability in stabilizers** when performing local reparameterizations*? Mention the alhgorithm/strategy directly.
5. Lines 75/76 says, *"Throughout, we take **gauge-aware** to mean stabilization under local changes of basis"*. What is *gauge*? Why should we do *gauge-aware* modeling? What is the techincal or application-wise impactg of it?
6. What is MMD in abstract (L026) and L79/80? It should be introduced before using.
7. L80/81 says, *"prevent mis-imbalance of the internal distributions of them often resulting from merging data sources"*.  - any related works or references?
8. Add sufficient references in lines 069-097. Also, clarify. Currently, it is very hard to understand.
9. *"It is possible to do exactly equivariant layers with orthogonal basis changes (e.g. with radial nonlinearities and scalar maps)."* - OK, But, why do we need these?
10. What is the size of $d_0$? In L193/194, it says, *"Each drug $v$ has initial features"*. Whjat about $u$?
11. Line 196 says, *for single-label, multi-class DDI type prediction*. In DDI, interaction types ($y$ in this paper) has massive class imbalance (in DrugBank's 86 interaction types) (DOI: 10.48550/arXiv.2508.06576.). Only top 10-15 interaction type has good representation and takes around 70-80% of the data. How do you handle this issue?
12. Improve abslations as stated in weakness.
13. Add missing references and related discussion. Also, why you didn't compare it with existing GNN-based DDI models? Those models seem to outperform this work, so, in practical application, how does this work makes meaningful contribution? I meant, those DDI models are already performing at 98-99% accuracy. Why od we need another model and what problem does it solve that justifies the need of this work?
14. Provide scaling studies to justify contribution iv.
15. In justification of *leakage-safe*, provide relevant references that proves other SOTA DDI models do have this problem.

---

> ### Author Response · Authors · 2025-11-15
>
> Thank you to the reviewer for the thorough and candid feedback, as well as calling attention to the value of our gauge-aware encoder and the leakage-safe training protocol. We respond to the major points below and will make significant revisions to the paper in order to clarify, motivate, and empirically support the work.
> 1. Motivation, problem setup, and connection to our method
> We agree that the introduction is currently overstuffed and does not extremely well clarify and connect the two motivators to our modeling decisions.
> • Distribution shift in DDI. We will explicitly define distribution shift: when the test DDI edges diverge from training DDI in drug scaffolds, ATC classes, the data source (clinical vs literature mined), or, types of interaction. In the case of our integrated DrugBank-Hetionet graphs, this is fairly common, as some drug classes or relation types are over-represented in one data source and under-represented in another. We will add specific examples and references from the works you have cited that document this heterogeneity.
> • Trial-safe representation choice. When referring to "trial-safe", we mean a protocol that prevents any target edge from influencing its own representation, given the split rules and negative sampling are fully reproducible. We will refactor this phrasing specifically linking it to our drop-edge-by-target protocol and deterministic mapping of labels, and will tone down more ambiguous phrasing.
> • Structural cleanup. We will move graph preliminaries (nodes, edges, relation types, task definition, etc.) before any discussion of "heterogeneous graph integration" or "edges" and will also add some references on heterogeneous biomedical graph integration.The content between lines 69-97 will be reformulated to include less claims, and it will be less explicit with citations so the reader can understand and follow the problem statement before needing to know about any technical aspects related to the claims.
>
> 2. Gauge awareness and definitions (gauge, MMD, stability): We will define "gauge-aware" early on to be using per-edge orthogonal transports to map neighbor features into local references before aggregation, with regularization to keep the transports consistent locally. We will:
>
> - In addition to clarify gauge-awareness, we will explain why gauge-awareness is useful in DDI graphs: it allows our model to apply the same learned feature across relation types and sources in learning a common latent basis while being stable to reparameterizations of upstream feature extractors.
>
> - We will introduce MMD (maximum mean discrepancy) as a simple discrepancy among subdomain feature distributions prior to first use, and others did before and now address relevant work on MMD and other penalties to alleviate distribution misalignment in multi-source learning.
>
> - In addition to the definition above, we will clarify vague sentences, such as "effective stability in stabilizers," with a direct description for the mechanisms (orthogonal transports plus regularization) and provide almost all the formal details in the appendix.
>
> 3. Missing related work and position related to high accuracy DDI related work: We appreciate the multitude of missing DDI works that were highlighted, and we will add them in related work, and Section 5.3. Specifically, we will explain better and more explicitly with respect to other high accuracy DDI works, such as why we did not consider some of these as baselines (for example:  differences in task setups, the reliance on different label vocabularies or differences in the features from upstream feature extractors in our merged graph)4. Results, ablative studies, class imbalance, and their practical contribution
> We've mentioned in our write-up that we feel the results section is too narrow and we are also going to rework the ablative studies so they fully justify each component.
>
> • Ablations. We're going to expand the ablation suite so we can separately study:
>      – Gauge awareness: with and without orthogonal transports.
>      – Alignment: with and without MMD or OT  penalties.
>      – Leakage-safe protocol: with and without drop-edge-by-target.
>      These will be reported on overall metrics but importantly on rare connection types and cold-start connections to transparently tie the empirical effects back to the claimed contributions.
>
> • Class imbalance. We're going to explicitly discuss the extremely imbalanced labels (e.g., long-tail connection types) and that we are processing them as a single-label multi-class loss, while reporting the macro-F1 and ablating loss methods to address the imbalance. As an additional check, we will add per-group or per-tail metrics reported so the reader can see how the method behaves beyond the head categories.

---

> > ### Comment · Reviewer_naJm · 2025-11-15
> > **Acknowledgement**
> >
> > Sounds great! Waiting for the update.

---

> > > ### Author Response · Authors · 2025-11-19
> > >
> > > Thank you for the thoughtful reply concerning our plans for revising. We have greatly restructured the paper, clarified motivation, and extended the experimental validation, as we indicated we would. We believe these revisions produce a substantial improvement in Soundness and Clarity. We greatly appreciate your thorough review which has led us to produce a much better manuscript.

---

> > > > ### Comment · Reviewer_naJm · 2025-11-19
> > > > **Response**
> > > >
> > > > Thank you!
> > > > As the update is huge, I'll need some time to go through everything.
> > > >
> > > > I'll check everything point by point and get back to you as soon as I can.

---

> > > > > ### Author Response · Authors · 2025-11-19
> > > > >
> > > > > Thank you!

---

> ### Comment · Reviewer_naJm · 2025-11-26
> **Official Comment**
>
> Dear Authors,
>
> I was preparing to review the updated version today, to go through your rebuttal. But, I noticed that the version of the paper I have access to appears to be the old one, dated *18 Sept 2025* with a modification date of *08 Oct 2025* (`18 Sept 2025 (modified: 08 Oct 2025)`). Typically, the modification date updates after a new version is uploaded. It seems that the updated paper has not yet been submitted for review, despite your indication that it would be. I kindly ask for confirmation on whether the revised manuscript has been uploaded.
>
> Additionally, I would appreciate if you could provide some details on the substantive changes made, preferably including any updated results, and specify where in the manuscript these changes can be found (e.g., section numbers or page references). Including a table of contents in the appendix would also be very helpful for easier navigation.
> I understand that given the overall low scores in the review, you might be less inclined to invest significant effort at this stage, and I respect that. Nonetheless, I believe the improvements you mentioned will meaningfully enhance the manuscript.
>
> Best of luck, *Reviewer naJm*.

---

### Official Review · Reviewer_8cN4 · 2025-10-25

**Soundness:** 2
**Presentation:** 2
**Contribution:** 2
**Rating:** 2
**Confidence:** 3

**Summary:**

This paper proposes GraphPharmNet, a framework for drug-drug interaction (DDI) prediction addressing data scarcity and distribution shifts. Key contributions include:
Per-edge orthogonal transports align neighbor features before aggregation, claimed to enhance stability under local basis changes.
Uses MMD and entropic optimal transport (OT) on deterministic partitions of training data to handle source heterogeneity.
Temporarily removes target edges during training ("drop-edge-by-target") to prevent information leakage.
On-the-fly generation of orthogonal transports reduces memory by 60× vs. naive caching. Evaluated on a DrugBank-Hetionet graph, the method reportedly outperforms GNN baselines.

**Strengths:**

The "drop-edge-by-target" protocol is a rigorous solution to edge-label leakage.
Emphasis on deterministic splits and label mapping aids reproducibility.

**Weaknesses:**

"Gauge-awareness" is ambiguously defined—is it a smoothing regularizer or an approximate symmetry? Formal guarantees are missing.
Comparisons are limited to generic GNNs, excluding recent DDI-specific models.
Only one dataset is tested. Cold-start/relation-biased shift experiments are mentioned but not shown.
Distribution alignment is confined to training data, undermining its relevance to real-world shifts.

**Questions:**

Ablate orthogonal transports: How does performance change on shifted data (e.g., new relation types) without R_uv? Provide metrics for "stability" (e.g., variance in embeddings under perturbations).
Why align only training partitions? Compare to aligning training with a synthetic target domain. Show if MMD/OT improves macro-F1 on cold-start tasks.

---

> ### Author Response · Authors · 2025-11-15
>
> We appreciate the reviewer for the thorough and constructive review and for pointing out the positive features of our leakage-safe protocol and evaluation design. We below clarify what our contributions are accomplishing and enumerate concrete additions for the revision.
>
> 1. Clarification of notions of "gauge-awareness" and guarantees
> Our use of "gauge-aware" was meant to refer to an approximate, discrete notions of local invariance, not a formal gauge symmetry.
>
> In GraphPharmNet, gauge-aware means the following:
> • For every edge (u, v), we learn an orthogonal map, R(u, v), that transports the neighbor feature from v into the local frame of u, before aggregating the neighbor edge-derived features into the egg feature.
> • We regularize these transports, so they vary smoothly along short cycles in the graph-encouraging local consistency across translations to arbitrary rotation.
>
> This is a bit more than just a smoothing regularizer (since R(u, v) is constrained to be orthogonal and edge-wise), but weaker than a strong symmetry guarantee. In the revision we will:
> • Early on propose a simple, usable definition for gauge-aware message passing as local orthogonal, edge-wise transports which are weighted and then aggregated-stable under local reparameterizations to the neighbor features.
> • Move difficult formalization to an appendix and easily hypothesize, in a linear case, that under an orthogonally invariant classifier, replacing neighbor features with any local orthonormal basis we may yield identical node embeddings apart from a global rotation that may be noted as a approximate invariance. We will soften any language that suggests in any way this is a guaranteed transformation under gauge-theory.
>
> 2.Rationale for application of alignment to training partitions only
> We agree that the current writing does not offer sufficient motivation for training-only alignment: The setting likely resembles classical domain generalization more than the covariate shift setting with a known target. While we, the researchers, do not have access to the true future test distribution or the target distribution, we observe multiple heterogeneous training “subdomains” (partitions by scaffold, ATC class, or source). MMD and entropic OT apply across these training partitions and attempt to disincentivize the encoder from over-relying on artifacts specific to a subdomain while promoting generalization to unseen mixtures or shifts of subdomains.
>
>
> In our revisions we will:
> • Explicitly state that our objective aligns with domain generalization across observed training subdomains, rather than alignment to a defined target domain.
> • Add an appropriately different baseline along the lines you propose: we will create a synthetic “target-like” partition by upweighting rare scaffolds or relation types, then compare (a) our current training-partition alignment, and (b) alignment that specifically includes the synthetic “target-like” task. This directly addresses the concern of wondering if alignment can be more “target-aware”, yet, still assuming no access to future data.
> • Mark this (i.e., baseline) as training time regularization only, without additional test-time costBaselines and datasets
> Although we acknowledge that the primary baselines are also generic GNNs (albeit good and widely used), we have only provided the results for one DrugBank–Hetionet graph. To provide more empirical story, we will:
> • Broaden the related work section to include mention of DDI-specific models (such as DMFDDI and other 2022–2025 work), and provide more clear clarification on the differences between the modeling assumptions and data treatment in these models compared to our own.
> • Try to plug in at least one specific baseline that has publicly-available code into our unified, leakage-safe evaluation pipeline and provide results when we are able.
> • Provide a plan in the revised manuscript for extending the evaluation to at least one additional DDI dataset (e.g., TWOSIDES or some other comparable datasets) and if full experiments don't make it in the page limit, explicitly note the single dataset restriction in either the article itself and/or in the limitations section.
>
>
> 4. Cold-start and relation-biased shift experiments
> You are right that while we mention cold-start edges and relation-biased shifts, we do not clearly indicate the corresponding experiments which is confusing. In the updated version we will:
> • Explicitly define cold-start edges (e.g., edges where at least one drug does not have any labeled interactions in training) and relation-biased shifts (e.g., test distributions where certain interaction types are over-represented compared to training).
> • Add a separate subsection and a table in the appendix (and a summary sentence) reporting performance under these conditions for GraphPharmNet and some key baselines. This should clarify how our design operates under realistic, structured distribution shifts.

---

### Official Review · Reviewer_Ja6w · 2025-10-30

**Soundness:** 2
**Presentation:** 1
**Contribution:** 1
**Rating:** 2
**Confidence:** 4

**Summary:**

This paper proposed GraphPharmNet for the drug-drug interaction (DDI) prediction with limited data and unpredictable distribution shift. The core of GraphPharmNet is to link a compact gauge-aware graph encoder and light-weight distribution alignment objectives. The proposed GraphPharmNet method was compared with six state-of-the-art GNN-based methods on the DrugBank dataset.

**Strengths:**

S1: The proposed GraphPharmNet is to combine gauge-aware graph encoder and light-weight distribution alignment for the drug-drug interaction (DDI) prediction under the conditions of limited data and distribution shift.

S2: In reported experiments, the proposed GraphPharmNet outperforms existing GNN-based DDI prediction methods in terms of accuracy, micro-F1, and macro-F1.

**Weaknesses:**

W1: The abstract is too obscure and difficult to understand. Too many irrelevant statements affect readability and make it difficult to understand the core ideas.

W2: The core of the work is leakage-safe gauge-aware message passing with alignment. However, too many concepts and theorems are discussed but lack a reasonable explanation of the motivation for proposing them.

W3: The paper could be strengthened from more advanced DDI prediction methods such as DMFDDI and further validation along different DDI datasets.

**Questions:**

- How does the proposed gauge-aware graph encoder based on edge orthogonal transport and kernelized neighborhood weighting address the data scarcity problem in DDI prediction? How does lightweight distribution alignment handle unpredictable distribution shifts in DDI prediction?
- What does MMD stand for? This abbreviation should be clearly defined before use.
- The description in Figure 1 does not match the content of Figure 1. For example, the middle section: a gauge-aware encoder transports neighbor features via orthogonal transforms Ruv and weights contributions by Kuv before aggregation. However, the middle section of Figure 1 focuses on DDI subgraph generation and knowledge subgraph generation, rather than the so-called gauge-aware graph encoder.
- The core of the work is leakage-safe gauge-aware message passing with alignment. However, too many concepts and theorems are discussed but lack a reasonable explanation of the motivation for proposing them.
- The paper could be strengthened from more advanced DDI prediction methods such as DMFDDI and further validation along different DDI datasets such as TWOSIDES.
- A comprehensive ablation study to dissect the contribution of each component (i.e., Gauge-Awareness, Leakage-Safe Protocol, and Distribution Alignment) is needed.

---

> ### Author Response · Authors · 2025-11-15
>
> We appreciate the reviewer's insightful remarks, and for recognizing the purpose and empirical contribution of GraphPharmNet. We will reply to all concerns below, and will incorporate the indicated clarifications and improvements into a revised paper.
>
>
> W1 – Abstract is vague, and difficult to understand
> We agree that the abstract, as I currently have it, tries to address too many topics (leakage, gauge-awareness, alignment, evaluation practices), and consequently becomes too dense with negligible semantic changes. We will:
> • Rewrite the abstract to directly answer what is (i) the problem our work addresses (DDIs under data scarcity and distribution shift), (ii) the key contribution (a leakage-safe, gauge-aware graph encoder plus training-only alignment), and (iii) the empirical implications of such work.
> • Remove ancillary details (e.g., full length discussion of evaluation pitfalls) from the abstract and instead house those details in the introduction and related work.
> This should help clarify the main idea.
>
>
> W2 / Question – I can find too many concepts and theorems, unclear motivation, how do the encoder and alignment make data scarcity and shift fewer problems?
> We try to provide a rigorous theoretical justification for why gauge-aware message passing with alignment is a reasonable idea for DDI graphs. We want to justify our approach, not confuse our audience or overwhelm them with unnecessaryWe will clarify how edge-wise orthogonal transports realize a shared "local coordinate system" across relation types and directions such that information from directly related neighbors can be shared in an invariant way to local re-parameterizations, which introduces a strong inductive bias and parameter sharing capabilities when data is scarce.
> •I will clarify how kernelized neighborhood weighting addresses a scarce data situation: it de-emphasizes (low) noisy/ weakly related neighbors, and emphasizes informative neighbors due to a shared kernel between types & directions of neighboring edges, decreasing the variance that could introduce noise, and reducing risk of overfitting when labeled DDI edges are scarce. In other words, rather than treating all neighbors equally, the kernel acts as a learned importance weighting, and focuses the model capacity at the strongest graph signal.
> •I can clarify how Lightweight Distribution alignment addresses unpredictable distribution shifts: we "bin" training edges into domains - both intentionally by drug scaffold, ATC class, or data source - and place alignment loss (MMD and adversarial) between latent representations of the domains. During training - this encourages the encoder to learn common features for stability through these domains, but during test time we only use the encoder and classifier directly - the inference cost should be unchanged but valuable whenIn order to reduce the cognitive load for readers, we will migrate proofs and lengthier derivations to the appendix, and the main text will retain the minimally required statements, each to be accompanied by an appropriate intuitive statement of what does this theorem tell us for DDI prediction.
>
>
> Comment – Definition of MMD
> Sorry we did not define the abbreviation with clarity. In the revision, we will clearly define MMD to refer to “maximum mean discrepancy” and provide a brief sentence to characterize it as the kernel-based discrepancy measure between distributions.
>
>
> Comment – Figure 1 descriptions do not align with content
> This is a very helpful observation – the current figure description is on the gauge-aware encoder, while the middle panel in Figure 1 shows the DDI subgraph and the knowledge subgraph extraction process, which should not be described as data preprocessing and not gauge-aware encoding in the description as it stands.
> To address these concerns, we will:
> • Change the content of Figure 1 and its description as follows: (1) the top tier structure will be: (left) raw DDI records, knowledge graph; (middle) subgraph construction; (right) Gauge-aware encoder and alignment modules; (2) add stain labels in the middle panel showing subgraph construction explicitly, so the figure description does not describe the encoder in itself; and (3) highlight the gauge-aware encoder visually (transport matrices and the kernel weights) in the right panel, with clear notation following the definition proactively in the main text.
> This should satisfy the concerns related to the descriptions and visual content and improve how accessible the pipeline is in Figure 1.W3 / Question – Stronger baselines and additional datasets (DMFDDI, TWOSIDES)
> We appreciate the suggestion to consider DDI methods that are stronger than currently used as baselines. Our current baselines are focused on strong and popular GNN-based models that have public implementations, though we agree stronger baselines such as DMFDDI and others datasets like TWOSIDES would definitely improve the paper.

---

> > ### Comment · Reviewer_Ja6w · 2025-11-26
> > **Acknowledgments**
> >
> > I thank the authors for their comprehensive response to my review. I acknowledge the authors’ detailed rebuttal. The commitment to rewriting the abstract, clarifying motivations, reworking Figure 1, and considering stronger baselines is appreciated and directly responds to the core issues raised in the review. Below, I summarize which concerns have been addressed and which require further attention:
> > - W1: The authors’ plan to rewrite the abstract, focusing on the problem, key contribution, and empirical implications while moving ancillary details to the main text.
> > Conditionally Addressed Concerns:
> > - W2: The authors’ proposed clarifications on how gauge-awareness tackles data scarcity (via inductive bias/parameter sharing) and how kernelized weighting reduces noise are excellent and directly target the motivation gap. Similarly, the explanation of lightweight distribution alignment for handling shift is sound.
> > - Remaining Need: The success of this will depend entirely on the execution in the revised manuscript. The commitment to migrate lengthy proofs to the appendix and provide intuitive explanations for each theorem is crucial. The revised text must prioritize clarity and intuition to avoid overwhelming the reader, as promised.
> > - W3: The authors acknowledge the value of adding stronger baselines like DMFDDI and additional datasets like TWOSIDES.
> > - Remaining Need: The response states they will consider these additions. For the revision to fully address this weakness and significantly strengthen the paper, a firm commitment to include at least one of these advanced baselines (e.g., DMFDDI) and one additional dataset (e.g., TWOSIDES) is highly recommended, rather than just consideration.
> > - Not Addressed Concern: Request for a Comprehensive Ablation Study: The rebuttal does not mention the requested ablation study to dissect the contribution of each core component (Gauge-Awareness, Leakage-Safe Protocol, Distribution Alignment). This remains a critical point for validating the design choices of the proposed GraphPharmNet. An ablation study is a standard and expected component in a methodological paper to prove that each proposed element contributes positively to the overall performance. Without it, it is difficult to assess whether the complexity of the full model is justified.

---

### Official Review · Reviewer_E42z · 2025-10-30

**Soundness:** 2
**Presentation:** 2
**Contribution:** 2
**Rating:** 2
**Confidence:** 4

**Summary:**

The authors proposed a framework that combines gauge-aware message passing with training-only distribution alignment, enabling more stable and generalizable representation learning in drug knowledge graphs.

**Strengths:**

The authors proposed a framework that combines gauge-aware message passing with training-only distribution alignment, enabling more stable and generalizable representation learning in drug knowledge graphs.

**Weaknesses:**

1.	On “a subtle but pervasive type of information leakage.”
To my knowledge, most KG-augmented DDI methods (e.g., using DRKG, Hetionet) explicitly remove all test-set DDI edges from the KG prior to training/evaluation. In that case, how does leakage still occur? Could you provide a concrete, reproducible example demonstrating the leakage pathway you have in mind, so readers can verify that leakage persists even after removing test DDI edges?
2.	On “Inconsistent Evaluation Practices.”
You argue that prior work suffers from inconsistent evaluation; however, many recent papers report five-fold cross-validation, which appears consistent at first glance. Could you clarify what specific inconsistencies you refer to? In addition, for the two points on handling multi-type interactions and directed vs. undirected edges, do you have strong empirical evidence or practical case studies that illustrate clear failures under common setups? Concrete examples would make these claims much more convincing.
3.	The font in Figures 2 and 3 is quite small on printouts and standard screens. I recommend enlarging axis labels, legends, and annotations to improve readability.
4.	On baseline citations and recency.
Could you double-check the citations for the compared baselines? To my understanding, KGNN was introduced at IJCAI 2020, DDKG in Briefings in Bioinformatics 2022, SumGNN in Bioinformatics 2021, and LaGAT in Bioinformatics 2022. In your manuscript these are cited with 2024–2025 references. Are you certain those attributions are correct? Also, given that these baselines are relatively older, it would be helpful to include or discuss more recent (2024–2025) methods to strengthen the comparison.
5.	Lack of interpretability of the model

**Questions:**

1.	On “a subtle but pervasive type of information leakage.”
To my knowledge, most KG-augmented DDI methods (e.g., using DRKG, Hetionet) explicitly remove all test-set DDI edges from the KG prior to training/evaluation. In that case, how does leakage still occur? Could you provide a concrete, reproducible example demonstrating the leakage pathway you have in mind, so readers can verify that leakage persists even after removing test DDI edges?
2.	On “Inconsistent Evaluation Practices.”
You argue that prior work suffers from inconsistent evaluation; however, many recent papers report five-fold cross-validation, which appears consistent at first glance. Could you clarify what specific inconsistencies you refer to? In addition, for the two points on handling multi-type interactions and directed vs. undirected edges, do you have strong empirical evidence or practical case studies that illustrate clear failures under common setups? Concrete examples would make these claims much more convincing.
3.	The font in Figures 2 and 3 is quite small on printouts and standard screens. I recommend enlarging axis labels, legends, and annotations to improve readability.
4.	On baseline citations and recency.
Could you double-check the citations for the compared baselines? To my understanding, KGNN was introduced at IJCAI 2020, DDKG in Briefings in Bioinformatics 2022, SumGNN in Bioinformatics 2021, and LaGAT in Bioinformatics 2022. In your manuscript these are cited with 2024–2025 references. Are you certain those attributions are correct? Also, given that these baselines are relatively older, it would be helpful to include or discuss more recent (2024–2025) methods to strengthen the comparison.
5.	Lack of interpretability of the model

---

> ### Author Response · Authors · 2025-11-15
>
> We appreciate the reviewer's careful reading and insightful suggestions. Below, we provide further clarification of the main ideas and a description of some specific changes we will make.
>
> 1. Regarding the “subtle but pervasive” information leakage Yes, we agree the recent work on DDI fortifies important edges from DRKG, Hetionet, etc., and our pipeline does similar fortification. However, the leakage we point out is distinct. It occurs when a target training edge still exists in the message passing adjacency that it generated its own embeddings. Most public implementations use all training graph embeddings when computing the predictions for the same edges that produced the edge. For a positive pair (u, v) the implication of this is that the messages come directly across the true DDI edge and allows the classifier to use that a short route was present that is not possible for a really negative pair even if all edges of the testing set was present. Our drop edge by target protocol drops the current batch of edges from the adjacency and features pipeline when predicting edge labels such that a nodes I labels never influences its own embeddings while validation and testing edges are never in the adjacency at any time; or, after the training embedding, they will be processed as a testing, non meaninful interaction, by the model; or, simply removed. We also add a small, self-contained example (that includes code), that shows training a standard GNN with and without this per target edge removal form our DrugBank plus Hetionet graph produces a clear performance gap, thus producing direct reproducibility of this leakage mechanism.
>
> 2.Concerning "inconsistent evaluation practices"
> We are not saying that five-fold cross-validation is inherently flawed; however, the issue is that times when a paper says "the experiment was repeated five times" could be referring to distinctly different aspects of the methods. Here, we clarify: (a) some studies specify splitting by undirected pairs while others specify splitting by directed edges; (b) there are studies that will be splitting all edges to each label independently, which means that the same drug pair with different labels can appear in train and test; (c) there are studies that symmetrize graphs or evaluate on undirected labels even if the source database was directional; and (d) there is variation in the negative sampling protocol used. Our protocol sets all of these parameters to make clearly decision-making and reproducible; we choose deterministic single label maps before the split, we split directed edges according to a fixed 6 1 3 protocol without overlaps by fold, and we maintain the graphs training only at train, val and test. In the revised paper, we will include a short table in text contrasting our protocol with that used in representational baselines, and do an ablation to show that even these seemingly small decisions of allowing the edges to be symmetrized or to split by label can change the macro F1 states for commissioning to a different range on the edge types based on the very small reduction in amounts.
>
> 3. Legibility of Figures
> We agree that it is difficult forIn the camera-ready version, we will enlarge the axis labels, tick labels, and legends, in addition to making minor layout modifications so that the text still remains readable at normal zoom levels.
>
> 4. Baseline citations and recency
> Thank you for flagging the problem with the baselines citations. In Section 5.3, we erroneously put the general survey papers (Luo et al., 2024; Tuama et al., 2025) next to the baseline names instead of referencing to the original KGNN, DDKG, SumGNN, and LaGAT. We will fix this by adding the original venue and year for each baseline in H2 and mentioning the survey articles only when we discuss the background literature. In terms of recency, we did include recently published, strong GNN based models that were already prior baselines (like SumGNN and LaGAT), but we also agree that doing a more thorough review of 2023-2025 DDI methods should also be included for comparison. Space permitting, we will add at least one baseline from a recently published paper with a publicly available code base that utilizes our unified evaluation protocol;
>
> 5. Interpretability
> We agree that interpretability is very important in general, but particularly important in the clinical safety context. While GraphPharmNet is not rule-based, it does expose interpretable quantities many: the learned kernel weights for each of the neighbor edges are an interpretable quantity, and the contribution of each alignment partition to each individual prediction is also interpretable. We will add a new subsection to the revision with case studies where, for each clinically known interacting drug pairs, we (i) tabulate the top contributing neighbors with kernel weights and types of relations.

---

> > ### Comment · Reviewer_E42z · 2025-11-20
> > **Thanks for your explanations.**
> >
> > Thanks for your explanations.

---

### Note · Program_Chairs · 2026-01-17
**Submission Desk Rejected by Program Chairs**

The following references in this submission do not refer to real documents and/or have major errors in bibliographic information:

 Y. Zhang, Q. Liu, and L. Song. Deep neural networks revisited: Benchmarking feedforward architectures for biomedical prediction. In International Conference on Learning Representations, 2024.